# Group Contextual Encoding for 3D Point Clouds

**Xu Liu**[1,2*] **Chengtao Li**[3] **Jian Wang**[4] **Jingbo Wang**[5] **Boxin Shi**[6†] **Xiaodong He**[2†]
[1]The University of Tokyo  [2]JD AI Research  [3]MIT  [4]Snap Inc.  [5]CUHK  [6]Peking University

## Abstract

Global context is crucial for 3D point cloud scene understanding tasks. In this work, we extend the contextual encoding layer that was originally designed for 2D tasks to 3D point cloud scenarios. The encoding layer learns a set of code words in feature space of the 3D point cloud to characterize the global semantic context, and then based on these code words, the method learns a global contextual descriptor to reweight the feature maps accordingly. Moreover, compared to 2D scenarios, data sparsity becomes a major issue in 3D point cloud scenarios, and the performance of contextual encoding quickly saturates when the number of code words increases. To mitigate this problem, we further propose a group contextual encoding method, which divides the channel into groups and then performs encoding on group-divided feature vectors. This method facilitates learning of global context in grouped subspace for 3D point clouds. We evaluate the effectiveness and generalizability of our method on three widely-studied 3D point cloud tasks. Experimental results have shown that the proposed method outperformed the VoteNet remarkably with 3 mAP on the benchmark of SUN-RGBD, with the metrics of mAP@0.25, and a much greater margin of 6.57 mAP on ScanNet with the metrics of mAP@0.5. Compared to the baseline of PointNet++, the proposed method leads to an accuracy of 86%, outperforming the baseline by 1.5%.

## 1 Introduction

Object detection in 3D point clouds is a challenging problem because it requires localizing and classifying objects from sparse and irregularly-distributed points. Conventional methods such as PointNet++ [16] and ASIS-PointNet++ [18] were proposed to solve this problem, which can learn the local features hierarchically. However, due to lack of global context modeling, the performance of PointNet [15] and PointNet++ [16] is limited. To resolve this issue, LG-PointNet++ [21] and PointWeb [27] proposed to model the global context by computing the pair-wise relations of points. However, their complexity is a quadratic function of the number of points, which is prohibitively expensive when dealing with large-scale point clouds.

On the other hand, for 2D Semantic Segmentation, Zhang et al., [24] proposed a contextual encoding layer to learn a descriptor to model the global context by encoding features with a dictionary with only a few code words and then aggregating the encoded information. In this paper, we extend the encoding approach to 3D point cloud, in which a dictionary containing only a few code words is learned to characterize the global semantic context, and then based on these code words, the method learns a global contextual descriptor to reweight the feature maps accordingly. Since the number of code words is constrained and much smaller than the number of data points, this method is computationally efficient. However, directly applying the encoding layer to 3D point clouds is inadequate. Compared to 2D scenarios, data sparsity becomes a major issue in 3D point cloud scenarios, and the performance of global contextual encoding quickly saturates when the number of

code words increases. To mitigate this problem, we further propose a Group Contextual Encoding (GCE) method, which divides the channel into groups and then performs encoding on group-divided feature vectors, to facilitate effective learning of global context in grouped subspaces for 3D point clouds.

We evaluate the effectiveness and generalizability of GCE-based method on three widely-studied 3D point cloud tasks. Experimental results have shown that the proposed method outperforms the VoteNet [13] with 3 mAP on the benchmark of SUN-RGBD [17], by the evaluation metrics of mAP@0.25, and a much greater margin of 6.57 mAP on a more challenging dataset of ScanNet [13], by a stricter evaluation metrics of mAP@0.5. We also demonstrate that our method can be generalized to other tasks like voxel labeling. Compared to the baseline of PointNet++ [16], the GCE layer leads to an accuracy of 86%, outperforming the baseline by 1.5%, which is the state-of-the-art performance on this benchmark.

To summarize, this work makes the contributions in the following aspects:

- We extend the contextual encoding layer to 3D point cloud scenarios to better model the global contextual information efficiently.
- We propose a group contextual encoding method dividing and encoding group-divided feature vectors to effectively learn global context in grouped subspaces for 3D point clouds.
- The proposed method shows better effectiveness and generalizability on multiple 3D benchmarks with state-of-the-art performance.

The source code[3] has been released to facilitate the reproduction of our results.

## 2   Related Works

**3D Object Detection.**   The existing paradigms for 3D detection on point clouds can be classified into three types: the Bird's Eye View (BEV) based methods, the Voxel based methods, and PointNet/PointNet++ based methods.

For BEV-based methods [3; 9; 12], the data are firstly projected on the ground plane with the bird's eye view and then the conventional convolution networks are applied to generate features and predict bounding boxes. For Voxel-based methods such as VoxelNet [28; 22; 10], the point clouds are firstly allocated to regular-sized grid in the 3D Cartesian space. Then the conventional 2D or 3D convolution neural networks are applied to extract features and predict bounding boxes. However, these methods inevitably introduce information loss at the initial pre-processing process, making them inadequate for scenes with cluttered points.

Recently, quantization-free PointNet-based detectors such as VoteNet [13], PointRCNN [11] and STD [23] are proposed. They can model the point cloud directly from the raw input with Point-Net/PointNet++ Backbone. Since errors in quantization/projection process can be avoided, these methods have achieved promising results on 3D objection detection benchmarks such as [17; 4].

**Context in 3D point clouds.**   To further improve the performance, modeling the contextual information is needed. The method in [21] is one of the pioneering works in obtaining global context for point cloud segmentation. But such global context is computed by scanning each pair or local patches exhaustively, which is computationally expensive for 3D indoor scenes.

**Dictionary Learning and Residual Encoding.**   Dictionary learning typically generates the code-words according to statistics of the feature descriptor, e.g., K-NN or K-means, in an unsupervised way. NetVLAD [2] and PointNetVLAD [1] encode features by aggregating the residual between the code words with hard/soft-assigned weights in an end-to-end manner. Zhang et al., [25; 24] revise this method by assigning the weight with residuals, so that the code words can be learnt from the distribution of descriptors. The code words, as well as scaling parameters of weights will be learned inherently by the network according to the loss function. The global context can also be computed by aggregating only a few number of code words in the dictionary, which keeps the computational cost affordable [24].

**Group Operations.**   One of the difficulties in dealing with 3D point cloud is data sparsity. To match the data sampling density in a 2D scenario, much more data points are needed in the 3D space.

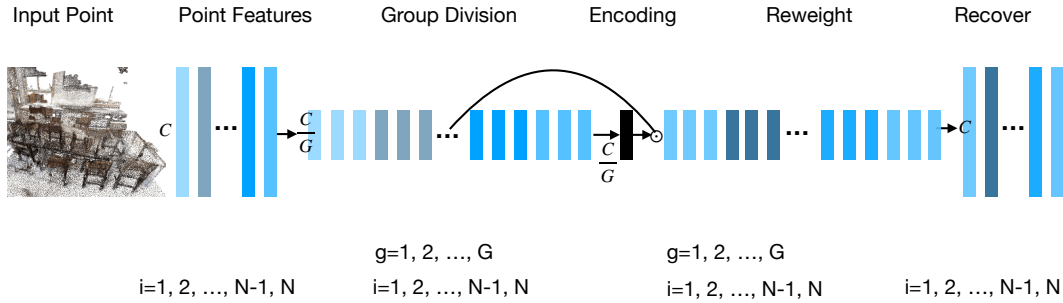

| Input Point | Point Features | Group Division | Encoding | Reweight | Recover |

g=1, 2, ..., G

i=1, 2, ..., N-1, N      i=1, 2, ..., N-1, N      i=1, 2, ..., N-1, N      i=1, 2, ..., N-1, N

Figure 1: Illustration of the proposed method. For features of $N$ input points, represented by vectors with index $i$, with the number of channels denoted as $C$, we first divide it into $G$ groups (in this case $G = 3$) and the subspace feature vector of the individual point can be represented by radix $g$. The group-divided features will pass through the encoding layer to generate the global descriptor and then multiply with it channel-wisely to reweight the features (the reweighting process is represented implicitly by darkened or lightened colors of the vectors). The reweighted grouped features will then be concatenated back to the original size.

In other words, compared to the 2D scenario, the sampling density is usually much sparser in 3D scenarios when the number of data samples are comparable. To mitigate this problem, we propose to learn global context in grouped subspace for 3D point clouds. Group convolution was originally introduced in AlexNet [8] to load the model in parallel for multiple GPUs. Group has also been regarded as an extra dimension for deep learning models, such as ResNeXt [20], ShuffleNet [26], and Group Normalization [19], and the channels are divided into groups. In spite of the similarity to these methods in form, we designate the grouping approach for the contextual information in 3D point clouds the first time.

## 3 Method

Our method can be viewed as an extension of PointNet++ by adding the component of Group Contextual Encoding (GCE). We first review the structure of PointNet++ and the concept of global context in Section 3.1 and then introduce the method to yield the global context with GCE in Section 3.2 and finally we propose our GCE-based PointNet++ in Section 3.3 .

### 3.1 Review of PointNet++ and 2D Encoding Layer

PointNet++ [16] is a typical "Encoder-Decoder" deep learning framework. In the encoding procedure, a series of Set Abstraction (SA) layers or Point Convolution layers are deployed to extract features hierarchically, and the number of points is reduced in this process. While in the decoding phase, the Feature Propagation (FP) decoding layers will be used to up-sample the features and recover the number of points back to the original input. The architecture can learn the local feature in a hierarchical manner but it lacks the ability to model the global context.

The encoding layer [24; 25] has originally been proposed for 2D scenarios. The prior information of context covering the entire image, referred to as global context, can serve as category clues to facilitate the scene understanding tasks, such as lowering the probability of the car appearing in the water. The global context is utilized as a global channel descriptor by encoding layer to reweight the channels. As a result, the useful information can be emphasized while the less meaningful one will be suppressed, which will lead to better performance. In the following, we will discuss how to acquire the global context of the 3D point clouds and utilize this information to re-calibrate the features.

## 3.2 Group Contextual Encoding Block

In this part, we will introduce our method to reweight the feature channels of point clouds with global context or global channel descriptor.

The "encoding layer" [24; 25], which is computed based on the residual between the input cloud and the learned code words can be leveraged for feature re-calibration. Since all points in the scene will contribute to the computing of the channel descriptor, therefore, this global channel descriptor can be attributed as "global context". The code words of [25; 24] are able to cover the problem space and learned in an end-to-end manner.

The channel-wise importance can be computed by encoding the residual between input point cloud features and the learned code words and then aggregating on these code words. However, due to data sparsity in the 3D point cloud scenario, overfitting is encountered when a large number of code words cover the problem space, which inhibits from gaining better performance by utilizing more code words. To mitigate this problem, we introduce the method of Group Contextual Encoding (GCE) Block to yield the global context for better feature representation.

Assume features of the point sets have a size of $N \times C$, where $N$ is the number of the points (a large number up to 1K to 20K in the scene) and $C$ is the number of channels which is a fixed small number and $C \ll N$. Additionally, channels of the point features are not entirely independent. Because the similar shapes, textures contribute to interdependent coefficients of the features. The point feature can therefore be represented by a small number of subspace vectors. In our design, the feature is divided into $G$ Groups and transformed into a size of $N \times G \times \frac{C}{G}$. The number of the points is equivalently augmented by $G$ times, which can help to address the issue of data sparsity and facilitate learning of global context in grouped subspace for 3D point cloud. The sub-space features of points can be represented by a set of independent vectors $d_K, K \ll N \times G$, which are learned in an end-to-end manner.

The feature of Point Convolution or Set Abstract (SA) layers will be multiplied in each channel with the global context. As a result, the point features will be empowered by the global contextual priors, which leads to a better performance. As shown in Figure 1, this is dubbed as the 3D Group Contextual Encoding block, or GCE block and this constitutes the our design of Group Contextual Encoding PointNet++ backbone.

**3D Group Contextual Encoding Layer.** To capture the global contextual feature of the point sets, we leverage the GCE layer to exploit statistics of the point sets. The encoding layer learns the code book inherently and then characterize the features according to the residual between the feature and the code words. The global contextual prior will then be multiplied channel-wisely with the original feature for re-calibration.

For the feature of the point sets, $X \in \mathbb{R}^{C \times N}$, $X = \{X_1, ..., X_N\}$, where $N$ is the size of the point sets and $C$ is number of the channels. We divide channels evenly into $G$ groups, which is illustrated in Figure 1. $G$ should be a positive divisor of $C$. The transformed feature can be represented by $\hat{X} \in \mathbb{R}^{\frac{C}{G} \times G \times N}$. The code book $D = \{d_1, d_2, ..., d_K\}$ has $K$ code words to be learned. The residual $r_{ijk} = \hat{x}_{i,j} - d_k$ will be weighted and summed by $e_k = \sum_{i=1}^{N} \sum_{j=1}^{G} e_{ijk}$.

$$e_{ijk} = \frac{e^{-s_k \|r_{ijk}\|^2}}{\sum_{k=1}^{K} e^{-s_k \|r_{ijk}\|^2}} r_{ijk}. \tag{1}$$

The scaling factor $s$ is a learnable parameter in the process. To obtain the global context of the scene, the information of each of individual encoders $e_k, k = \{1, 2, ..., K\}$ will pass through a combination of Batch Normalization [7] and ReLU operations, denoted with $\phi$, then aggregated with "sum" operation. This procedure is described using Equation 2.

$$e = \sum_{k=1}^{K} \phi(e_k). \tag{2}$$

The computation complexity of whole procedure is $O(N \times G \times K)$. Since $K$ and $G$ are small integer values, this method is computationally economical.

To exploit the aggregated information $e$, it will pass through a fully connected layer and sigmoid activation function and be utilized as channel attention, which is similar to [6]. This process is given

by $\gamma = \sigma(We)$, where the $\sigma$ denotes for the sigmoid activation, and $W$ stands for the weight of the FC (Fully Connected) layer. The global descriptor $\gamma$ will be multiplied in each channel with the input feature $\hat{X}$ with the channel wise multiplication: $Y = \hat{X} \odot \gamma$ as the modulator of the feature-map. Which is similar to the method mentioned in SENet [6]. The feature map will be concatenated back to the size of $C \times N$ finally, as shown in Figure 1.

**Relation with prior methods.** It should also be noted that when $G = 1$, this operation is degenerated into the encoding layer [24; 25]. This method computes the global context point-wisely and fails to address the issue of data-sparsity.

We follow EncNet [24] to use $G = 1$ and $K = 0$ to denote the "global average pooling" introduced in SENet [6]. Though this method performs well for the regular-sized and rectangular-shaped 2D images where the global information can be aggregated on $1 \times 1$ sized centroid, it is not effective for the 3D point clouds with irregular shape and distribution, as shown by our result in Section 4.

### 3.3 Group Contextual Encoding PointNet++

The GCE Block can be integrated into a variety of existing point cloud deep learning models. For example, to extend the original PointNet++ [16] with global context, we build a GCE-PointNet++. The configuration and architecture will be given in the supplementary material. Compared with vanilla PointNet++ [16], GCE Block introduced in Section 3.3 replaces the original SA layers as the building block. For each layer in the encoding phase, the feature will be improved by the global context aggregated from the feature of the current stage. And we empirically choose the code word number $K = 8$ for each of the layers. More details of the configuration can be found in Section 4 and supplementary material. In contrast with the original PointNet++ [16], the feature from each of the Point Convolution layers will be integrated with the global context. As a result, better performance can be achieved.

## 4 Experiment

### 4.1 3D Object Detection in Point Clouds

To verify the efficacy of Group Contextual PointNet++, we experiment on VoteNet [13] while use PointNet++ [16] as the baseline. The model is deployed on the benchmarks of SUN-RGBD [17] and ScanNet [4]. In experiments, we follow the same protocol in [13] and use the metrics, mean average precision (mAP), at IoU threshold of $0.25$ for evaluation.

**Dataset.** SUN RGB-D [17] for 3D indoor scene understanding consists of around $10K$ RGB-D images annotated with $64,595$ oriented 3D bounding boxes for nearly $40$ object categories. In our experiment, following [13] we split the training/testing set and report 3D detection performance on the 10 most common categories.

ScanNet [4] provides a wider range of indoor scenes with more densely scanned objects compared with the SUN RGB-D dataset. We use the $1205$ scans for training and $312$ scans for testing, respectively. Vertices from meshes are sampled as the input point clouds. Following the ground truth annotation mentioned in [13], we predict axis-aligned 3D bounding boxes in these scenarios.

#### 4.1.1 Ablation Studies

We set a series of ablation studies to investigate the components of our methods.

The seed layers of the PointNet++ [16] are dubbed with "SA1,SA2,...,FP1,FP2,...". To show the difference from the original PointNet++, for instance, we use the item "SA2'" to represent the second layer of Group Contextual Encoding Blocks, so is SA3', SA4'. The FP (Feature Propagation) operators are kept the same with PointNet++. Therefore, the names of FP layers are unchanged.

**Code word number $K$.** To verify the choice of code word number $K$ in the dictionary, we conducted experiment on SA2' with a series of numbers $K = 0, 8, 16, 24, 32$. We also choose $G = 1$, which is the case of "encoding layer" [25; 24] and the results are shown in Table 1.

It can be shown that the improvement of global average pooling ($K = 0, G = 1$) is limited, only $0.5$ mAP improvement on ScanNet V2 compared with the baseline method of SA2. The global

Table 1: Ablation studies of code word number $K$ with SA2$'$ feature of Group Contextual Encoding PointNet++ and G is set to be 1. Evaluated with mAP@0.25.

| Seed Layer | SUN RGB-D V1 | ScanNet V2 |
|---|---|---|
| SA2 | 51.2 | 51.2 |
| SA2$'$ ($K = 0$, $G = 1$) | 51.9 | 51.7 |
| SA2$'$ ($K = 8$, $G = 1$) | 54.6 | 53.0 |
| SA2$'$ ($K = 16$, $G = 1$) | 54.7 | 53.1 |
| **SA2$'$** ($K = 24$, $G = 1$) | **55.5** | **53.5** |
| SA2$'$ ($K = 32$, $G = 1$) | 55.0 | 53.4 |

Table 2: Ablation studies of Channel Number, Encoding layer and Grouping method on SUN-RGBD benchmark. The w/o encoding refers to the cases without encoding, the result of $G = 1$, noted as w/o group division and the result of our method, noted as w/ group division as well as "channel shuffle", noted as "w/ shuffle" are also listed for comparisons.

| SA2/SA2$'$ layer | w/o encoding | w/o group division | w/ group division | w/ shuffle |
|---|---|---|---|---|
| $C \times 1$ | 51.2 | 54.6 | **55.4** | 54.4 |
| $C \times 2$ | 54.0 | 55.2 | **56.8** | 56.1 |
| $C \times 3$ | 54.2 | 55.8 | **57.1** | 56.3 |

average pooling widely used for 2D tasks is not effective for the 3D point clouds. This paradox is easy to explain: unlike 2D images, the distribution and shape of 3D point clouds is irregular, thus the simple operator of global average pooling is unable to provide a true description for the global scene. Therefore, it does not perform well for irregular 3D point clouds.

In theory, the performance will increase with the code number $K$ based on the assumption that complex scenes can be better characterized by exploiting more independent code words. In practice, we find the performance gets saturated when $K$ increases from 8 to 32. The encoding layer [25; 24] can yield only limited improvement by utilizing more code words. For example, on the benchmark of ScanNet, it can only yield limited improvement of 0.5 mAP when $K$ increases from 8 to 32 (Though the improvement will be slightly higher on SUN-RGBD). The curve in Figure 2 also illustrates this trend of saturation. This is due to the "sparsity" of the point features, 8 code words are enough to represent the entire scene, therefore we choose $K = 8$.

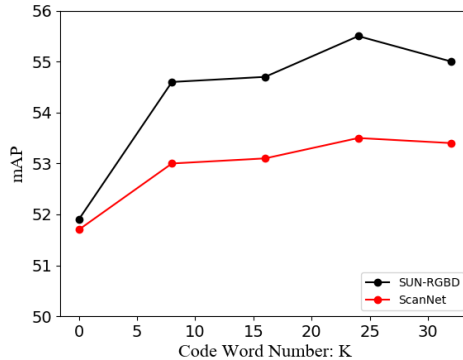

Figure 2: The performance of SA2$'$ layer w.r.t the code word number $K$ on SUN-RGBD and ScanNet datasets. $G$ is set to be 1.

For the following part, we will introduce the method to boost the performance independent of the code words by exploiting "groups".

**Channel Number.** Since the channel will be divided into groups, the effect of this variable needs to be investigated independently. We conduct the experiments of multiplying the channel number without incorporating the GCE Block, noted as w/o encoding, the details of the configuration can be found in the supplementary material. The results in Table 2 show that the performance increases with a margin of 2.8 mAP and 3.2 mAP when the channel number is multiplied by $\times 2$ and $\times 3$, denoted as $C \times 2$ and $C \times 3$, respectively. Since the performance gets saturated when the number of channels is multiplied by $\times 3$, we choose $C \times 3$ as the default setting in the following experiments for optimal performance.

**Comparison with Encoding layer.** We also make comparison of performance with encoding layer [24; 25], denoted by $G = 1$ or "w/o group division" in Table 2, and our method is denoted by "w/ group division" in Table 2. Even though the encoding layer will improve the performance, for example, it yields an increase of 1.6 mAP for the baseline method of "$C \times 3$", but it is only limited

Table 3: Comparison with the state-of-the-art algorithm on SUN RGB-D V1 benchmark.

| Method | Input | bathtub | bed | book shelf | chair | desk | dresser | nightstand | sofa | table | toilet | mAP |
|---|---|---|---|---|---|---|---|---|---|---|---|---|
| F-PointNet [14] | Geo+RGB | 43.3 | 81.1 | 33.3 | 64.2 | 24.7 | 32.0 | 58.1 | 61.1 | 51.1 | **90.9** | 54.0 |
| VoteNet [13] | Geo Only | 74.4 | 83.0 | 28.8 | **75.3** | 22.0 | 29.8 | 62.2 | 64.0 | 47.3 | 90.1 | 57.7 |
| **Ours** | Geo Only | **78.1** | **85.9** | **33.7** | 74.7 | **27.7** | **35.7** | **65.7** | **66.4** | **51.4** | 87.7 | **60.7** |

success. Our method of "w/ group division" with the choice of $G = 12$ shows an extra boost of 1.3 mAP in performance. Our method is advantageous over the encoding layer because we can resolve the data-sparsity issue and facilitate the learning of global context. We choose "$C \times 3$, $G = 12$, $K = 8$" as the default setting in the following experiment. More discussion on the choice of "$G$" can be found in the supplementary material.

**Grouping Methods.** In the experiment, the "Grouping" method follows the rule of "locality" that for feature of each individual point, the vectors with adjacent channels will be grouped together. We also compare with the method of "shuffle" [26], denoted by "w/ shuffle", which can weaken such constraint of "locality". But the results given in Table 2 show that no significant performance improvement is gained when "channel shuffle" is incorporated.

**Comparison with SA layer.** Table 2 also shows the improvement of our methods compared with the original SA layer of PointNet++ [16]. For example, for the setting of $C \times 3$, our method has outperformed the original SA layer with **2.9** mAP. More ablation experiments on different seed layers will be given in the supplementary material.

As discussed above, we choose "$K = 8$, $C \times 3$, $G = 12$" as the default setting for our experiments in the following sections.

### 4.1.2 Main Results

In this part, we compare our results with the previous state-of-the-art methods, including VoteNet [13] and F-PointNet [14], the results show that our methods have outperformed state-of-the-arts methods on SUN-RGBD [17] and ScanNet [4] benchmarks by a large margin. The visualization of the object detection results can be found in Figure 3.

The result on SUN-RGBD benchmark can be found in Table 3. Compared with the F-PointNet, our performance of **60.7 mAP** has surpassed it with **6.7 mAP**. It should be noted that it is not a fair comparison for us because the F-PointNet [14] has utilized an additional RGB image as input while we only use the geometry information as input. Compared with the "Geometry Only" method of

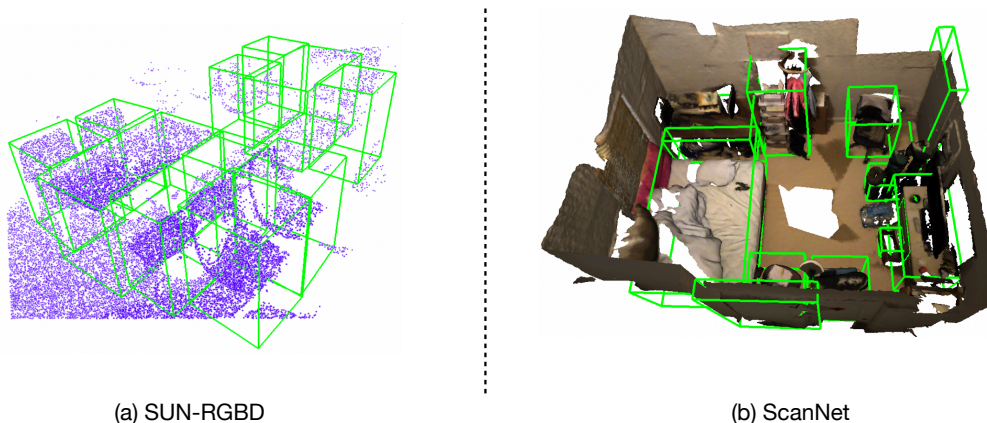

(a) SUN-RGBD        (b) ScanNet

Figure 3: Visualization of 3D Detection on SUB-RGBD (a) and ScanNet (b).

Table 4: Comparison of our method with state-of-the-art methods on ScanNetV2, evaluated with mAP@0.25.

| | cab | bed | chair | sofa | table | door | wind | bkshf | pic | cntr | desk | curt | fridg | showr | toil | sink | bath | ofurn | mAP |
|---|---|---|---|---|---|---|---|---|---|---|---|---|---|---|---|---|---|---|---|
| 3DSIS Geo [5] | 12.75 | 63.14 | 65.98 | 46.33 | 26.91 | 7.95 | 2.79 | 2.3 | 0.00 | 6.92 | 33.34 | 2.47 | 10.42 | 12.17 | 74.51 | 22.87 | 58.66 | 7.05 | 25.36 |
| VoteNet [13] | 36.27 | 87.92 | **88.71** | **89.62** | 58.77 | 47.32 | 38.10 | 44.62 | 7.83 | 56.13 | **71.69** | 47.23 | 45.37 | 57.13 | 94.94 | 54.70 | **92.11** | 37.20 | 58.65 |
| **Ours** | **38.20** | **88.34** | 87.13 | 84.48 | **65.24** | **48.14** | **41.40** | **48.00** | **8.10** | **60.95** | 70.17 | **47.59** | **46.59** | **73.60** | **98.36** | **59.38** | 88.23 | **40.98** | **60.83** |

Table 5: Comparison of our method with state-of-the-art methods on ScanNetV2, evaluated with mAP@0.5.

| | cab | bed | chair | sofa | table | door | wind | bkshf | pic | cntr | desk | curt | fridg | showr | toil | sink | bath | ofurn | mAP |
|---|---|---|---|---|---|---|---|---|---|---|---|---|---|---|---|---|---|---|---|
| 3DSIS Geo [5] | 5.06 | 42.19 | 50.11 | 31.75 | 15.12 | 1.38 | 0.00 | 1.44 | 0.00 | 0.00 | 13.66 | 0.00 | 2.63 | 3.00 | 56.75 | 8.68 | 28.52 | 2.55 | 14.60 |
| VoteNet [13] | 8.07 | 76.06 | 67.23 | 68.82 | 42.36 | 15.34 | 6.43 | 28.00 | 1.25 | 9.52 | 37.52 | 11.55 | 27.80 | 9.96 | 86.53 | 16.76 | 78.87 | 11.69 | 33.54 |
| **Ours** | **8.25** | **81.41** | **70.36** | **70.71** | **48.13** | **17.63** | **17.54** | **41.30** | **2.88** | **30.73** | **44.62** | **15.41** | **29.72** | **29.68** | **88.79** | **25.06** | **82.87** | **19.61** | **40.11** |

VoteNet [13], the performance of our method has succeeded it with **3 mAP**, on a shallower layer of FP1 and fewer point numbers compared with VoteNet. The discussions on choosing this seed layer are presented in the supplementary material.

As shown in Table 4, our method has achieved the performance of **60.8 mAP** on the benchmark of ScanNet [4]. This performance has exceeded the original VoteNet [13] with 2.2 mAP. It may seem to be only a slight improvement, while the evaluation on a stricter metrics, mAP @ IoU 0.5, illustrated in Table 5 shows that compared with the performance of original VoteNet [13], our method has outperformed it significantly with **6.57 mAP** and reaches the performance of **40.11 mAP** on this benchmark, which demonstrates the efficacy of GCE Block.

## 4.2   ScanNet Semantic Voxel Labeling

We evaluate our method on the task of Scan-Net voxel labeling. We do not incorporate RGB information, and we follow the same pre-processing technique, training protocol and evaluation method used in [16] for a fair comparison. In this experiment, we adopt the setting of $K = 8$, $C \times 3$, $G = 12$ and integrate it with PointNet++ [16]. The details of the experiment are introduced in the supplementary materials and the results are reported in Table 6.

Table 6: ScanNet Voxel Labeling Performance.

| Method | Accuracy % |
|---|---|
| PointNet++ [16] | 84.5 |
| PointCNN [11] | 85.1 |
| LG-PointNet++ [21] | 85.3 |
| PointWeb [27] | 85.9 |
| **Ours** | **86.0** |

The results show that our result has outperformed the baseline method of PointNet++ [16] with an increase of 1.5% in accuracy. It should also be noted that our method has outperformed other state-of-the-art methods, including PointCNN [11], LG-PointNet++ [21], and PointWeb [27].

## 5   Conclusions

We have presented Group Contextual Encoding as an effective method to acquire the global context in 3D point clouds, and evaluated this method on several prevailing benchmarks of 3D point clouds. Experimental results have shown that the proposed method outperforms the non-grouping baseline methods significantly across the board, and demonstrates state-of-the-art performance on these benchmarks, indicating our method as a compelling alternative to the original "encoding layer" for global context in 3D Point Clouds.

## Broader Impact

Our "Group Contextual Encoding" can be directly applied to the 3D point cloud scene understanding tasks including 3D object detection, voxel labeling, and segmentation. Our research can also support downstream research and applications such as autonomous driving, robotics, and AR/MR. We will investigate the generalizability of our method to other tasks and frameworks, e.g., Graph Convolution network, 3D sparse CNNs, where the global context plays a crucial role in these tasks.

On the other hand, this technology may also endanger the employment of human servants and drivers because they may be replaced by autonomous robots and vehicles, which may cause the potential social problems. This issue should be taken seriously and measures should be taken for preparation.

## Acknowledgments

This research was supported by GCL program of The Univ. of Tokyo by MEXT and in part by National Natural Science Foundation of China under Grant No. 61872012, National Key R&D Program of China (2019YFF0302902), and Beijing Academy of Artificial Intelligence (BAAI). We also thank Dr. Jing Huang and Dr. Youzheng Wu of JD AI Research for logistical and technical support during the research.

## Footnotes

*This work is done in JD AI Research

†Corresponding authors: shiboxin@pku.edu.cn, xiaodong.he@jd.com

[3] https://github.com/AsahiLiu/PointDetectron

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
