[Supplementary Material]

# Supplementary Materials

**Xu Liu**[1,2*] **Chengtao Li**[3] **Jian Wang**[4] **Jingbo Wang**[5] **Boxin Shi**[6†] **Xiaodong He**[2†]
[1]The University of Tokyo  [2]JD AI Research  [3]MIT  [4]Snap Inc.  [5]CUHK  [6]Peking University

## A  Overview

This supplementary material provides the details of the experiment in the paper. We introduce the details of 3D object detection in Section B and details of ScanNet voxel labeling in Section C.

## B  The Experiment on VoteNet

We introduce the implementation details and additional ablation studies of 3D detection in this part.

### B.1  Implementation Details

**Architecture.** We adopt the framework of VoteNet [2], which can be divided into three parts. The backbone, voting and clustering module, and proposal module. Only the backbone is replaced with our method of Group Contextual Encoding PointNet++ (GCE PointNet++) in our experiment.

The configuration of the GCE PointNet++ is shown in Table 1. The numbers are explained as follows. The GCE layer has a receptive field determined by radius $r$, MLP network of $MLP[c_1, ..., c_k]$ and $n$ subsampled points. These parameters are inherited from SA layers. Additionally, we use $K$ to represent the number of code words and $G$ to represent the number of groups in the GCE Block. In short, the GCE layer can be characterized by $(n, r, K, G, [c_1, ..., c_k])$. It should also be noticed that the number of $c_k$ is multiplied 3 times in Table 1, which refers to the "$C \times 3$" in our experiment. We can change the expression of "$\times 3$" in the table to "$\times 2$" and "$\times 1$" to get the configuration of $C \times 2$ and $C \times 1$ respectively.

Feature Propagation (FP) layers upsample the input point sets to output point set via interpolation and then pass the feature through MLP layers specified by $[c_1, ..., c_k]$

Table 1: The configuration of GCE PointNet++ in our experiment of 3D Detection.

| Layer Name | Input Layer | Type | Output Size | Layer Params |
|---|---|---|---|---|
| SA1′ | Raw Input | GCE | (2048,3+128×3 ) | (2048, 0.2, 8, 12, [64, 64, 128×3]) |
| SA2′ | SA1′ | GCE | (1024, 3+256×3) | (1024, 0.4, 8, 12, [128,128,256×3]) |
| SA3′ | SA2′ | GCE | (512, 3+256×3) | (512, 0.8, 8, 12, [128,128,256×3]) |
| SA4′ | SA3′ | GCE | (256, 3+256×3) | (256, 1.2, 8, 12, [128,128,256×3]) |
| FP1 | SA3′, SA4′ | FP | (512, 3+256×3) | [256,256×3] |
| FP2 | SA2′, SA3′ | FP | (1024, 3+256×3) | [256,256×3] |

**Training and Inference.** We adopt the same data augmentation methods with VoteNet [2] . Here we also adopted the same optimizer, Adam Optimizer [1], which is utilized with an initial learning rate 0.001. Learning rate is scheduled to be decayed by the factor of 0.1 after 80 epochs and another

0.1 after 120 epochs. There are 180 epochs in total, which is the same with VoteNet[2]. The whole model is trained on a single Nvidia Titan-X GPU.

During inference, the points of the entire scene are taken as the input. With a *single shot pass*, the region proposals are generated by the framework and further post-processed by 3D NMS method.

## B.2 Additional Ablation Studies

**Group Number** $G$**.** We investigate the performance w.r.t the group number $G$ on the dataset of SUN-RGBD v1. The $G$ should be an divisor of $C$ and the results are illustrated in Table 2. The items of the first row of $C \times 1$ , means the channel number is unchanged, has revealed that when $G$ is small, for instance, $G = 2$, the performance is close to encoding layer [6; 5], or $G = 1$. When $G$ is too large, the Channel per group will be reduced, the improvements by group division will be then dropped. And the optimal $G$ or defined as $G^*$ will be an number between $1$ and $C$ and in this case is $4$.

We also conducted experiments by increasing the output channel $2\times$ and $3\times$, denoted by $C \times 2$ and $C \times 3$ in Table 2. It should be noted that $12$ is indivisible by $C \times 2$ and $C \times 1$, therefore these items are blank in the table.

The result shows that the optimal choice of $G$ grows in linear relationship with $C$. For example, when channel number is unchanged, the $G^* = 4$, and this value is $8, 12$ when the channel number is multiplied $2\times$ and $3\times$ respectively. In this experiment, we choose "$C \times 3, G = 12$ as the default setting.

Table 2: Ablation studies of Group Number and Channel factor on Sun RGB-D V1, $K$ is set to be $8$.

| $G$ | 1 | 2 | 4 | 8 | 12 | 16 |
|---|---|---|---|---|---|---|
| $C \times 1$ | 54.6 | 54.9 | **55.4** | 54.6 | _ | 54.9 |
| $C \times 2$ | 55.2 | 55.5 | 55.8 | **56.8** | _ | 55.4 |
| $C \times 3$ | 55.8 | 55.8 | 55.4 | 56.7 | **57.1** | 57.0 |

**More results w.r.t.** $K$ **and** $G$**.** The performance of the original Encoding layer ($G = 1$) will saturate quickly with the code words. However, the results in the Table 3 show that our method ($C \times 3$, $G = 2$ and $C \times 3$, $G = 4$) can lead to the increase on accuracy without saturation when the number of code words is increased up to $32$.

Table 3: Ablation studies of SA2$'$ layer w.r.t. $G$ and $K$ on Sun RGB-D V1. $C$ is fixed to be $C \times 3$.

| $K$ | 8 | 16 | 24 | 32 |
|---|---|---|---|---|
| $C \times 3, G = 1$ | 55.8 | 55.5 | **56.2** | 55.4 |
| $C \times 3, G = 2$ | 55.8 | 56.2 | 56.4 | **56.7** |
| $C \times 3, G = 4$ | 55.4 | 55.6 | 56.3 | **56.6** |

**The performance on different seed layers.** Similar to Table 8 of VoteNet [2], we also showed the performance of different seed layers for the benchmark of SUN-RGBD and ScanNet in Table 4 and in Table 5 respectively. We can infer from these results that the GCE block can improve the performance significantly on these benchmarks.

On the benchmark of SUN-RGBD, we found that the performance of FP2 layer is less satisfying than FP1 layer. Similar result is also shown in the original VoteNet [2] that the performance of FP2 layer is better than FP3 layer, implying FP operation is not an optimal choice for decoding layer. The methods to design a suitable decoding layer for point convolution could be a future research topic.

Table 4: Ablation studies of PointNet++ and our module with different seed layers, evaluated on SUN-RGBD

| Seed Layer | SA2/SA2$'$ | SA3/SA3$'$ | SA4/SA4$'$ | FP1 | FP2 |
|---|---|---|---|---|---|
| PointNet++ | 51.2 | 56.3 | 55.1 | 56.6 | **57.7** |
| **Ours** | 57.1 | 58.0 | 59.3 | **60.7** | 59.1 |
| $\Delta$ | **5.9** | 1.7 | 4.2 | 4.1 | 1.4 |

Table 5: Ablation studies of PointNet++ and our module with different seed layers, evaluated on ScanNet.

| Seed Layer | SA2/SA2$'$ | SA3/SA3$'$ | SA4/SA4$'$ | FP1 | FP2 |
|---|---|---|---|---|---|
| PointNet++ | 51.2 | 54.3 | 47.4 | 56.6 | **58.6** |
| **Ours** | 56.3 | 58.3 | 53.9 | 59.0 | **60.8** |
| $\Delta$ | 5.1 | 4.0 | **6.5** | 2.4 | 2.2 |

## C   Experimental Details on ScanNet Voxel Labeling

In the experiment, we followed the previous data processing methods [3; 4], the points are uniformly sampled and divided into the block with the size of $1.5m \times 1.5m$. There are 8192 points sampled on-the-fly during the training process. The architecture is built upon the Pointnet++ [4], we replace the SA modules with GCE blocks and choose $K = 8, G = 12, C \times 3$ as the default setting.

## Footnotes

*This work is done in JD AI Research

†Corresponding authors: shiboxin@pku.edu.cn, xiaodong.he@jd.com