[Reviews · NeurIPS 2020]

Review 1

Summary and Contributions: This paper extends the idea of a "global context", which is a list of learned codewords, to 3D point cloud analysis. A PointNet++ is adapted by splitting the intermediate features into equally sized groups and by using encoding layers to learn codewords. The proposed network is evaluated in ablation studies, verifying the parameters and the influence of the proposed steps, and in comparisons with prior art on a large state-of-the-art dataset, showing very good results.

Strengths: + The evaluation is thorough and convincing. Ablation studies verify the method's building blocks, a large state-of-the-art dataset is used, the method is compared against state-of-the-art methods. + The results are very good and exceed prior art. + The method's steps are well motivated and explained + 3D Deep Learning is still a very active field that is far from mature or saturated, making the contribution relevant to the audience.

Weaknesses: - Overall, it is strongly recommended that the authors have the paper proof-read. There are numerous minor grammatical / syntactical hickups that make it difficult to follow the flow of the paper. Some examples are below in "Additional feedback", though that is not a complete list. - The evaluation is missing crucial details about the training of the network (optimizer, parameters), which might make it more difficult to reproduce the results. - The theoretical novelty of the approach is somewhat limited. The main contribution is the usage of encoding layers from [23] in a PointNet++ CNN. However, that step is well motivated and yields good results.

Correctness: The method is well described, the evaluation is correctly performed.

Clarity: The writing could and should be improved (see "Weaknesses", see "Additional feedback"). The method can be well understood from the paper, but I believe the description could be improved to make understanding it even easier -- both in terms of how the description is structured, and by improving grammatical issues.

Relation to Prior Work: Yes. The extensions and differences to PointNet++ are described, as are the papers that provided the ideas for those extensions ([23]).

Reproducibility: Yes

Additional Feedback: - Chapter 3.2 describes both the splitting of the feature vectors C (l.137--145) as well as the encoding layers. I'd recommend to put those into separate paragraphs to better guide the reader. - l.174--181: I like this analysis very much. It allows to better categorize the presented method. - l.141: I do not quite understand the reasoning for splitting up the vector. It apparently works, but maybe you can describe the initial motivation in more detail, or reference a paper where this was done before. - l.158: Shouldn't C be divisible by G (not G by C)? - l.42: Typo "effectivelearning" -> "effective learning" - l.75: Typo "liuxu436" - l.104: "section3.1" -> "section 3.1" - l.127: "space and learned" -> "space and are learned" - l.132: "words to cover" -> "words cover" - l.151: "sets. We" -> "sets, we" - l.156: "number of the point sets" -> "size of the point set", "number of the channel" -> "number of channels"? == After Rebuttal and Discussions == Though the novelty is somewhat limited, as a combination of previous ideas, the experiments are well executed and show very promising results. I believe the the global context has a significant potential, as it is potentially able to aggregate information from different regions in the input space, allowing a better overall reasoning of the network.


Review 2

Summary and Contributions: This works aims to improve existing architectures for 3D point cloud analysis by incorporating global context into the computed features, adapting an approach originally proposed for images in [23]. It demonstrates improved performance for 3D object detection as compared to VoteNet [12] on SUN-RGBD and ScanNet data, as well as compared to PointNet++ on a voxel labeling task.

Strengths: The main technical contribution of the paper is to adapt the codeword approach of [23] to the sparse setting of point cloud data.They introduce a block called the "Group Contextual Encoding Layer" (GCE) that avoids the overfitting encountered with a large number of codewords by partitioning the feature channels into groups and doing a codebook on a per group basis. A positive aspect of the paper is the careful ablation studies provided.

Weaknesses: The adaptation of the contextual image encoding from [23] is rather straightforward. It seems that *how* the channels are grouped can also matter for the quality of the learned context -- but the paper does not discuss this issue and only groups the channels according to their order of appearance in the future vector. This reviewer appreciates the careful rebuttal by the authors, where many important detailed are clarified about the network implementation and the comparative studies,as well as relations to prior work. Based on that, this reviewer has raised the score to a 7.

Correctness: Yes, the paper seems correct.

Clarity: Reasonably so -- the exposition is decent.

Relation to Prior Work: Yes.

Reproducibility: Yes

Additional Feedback:


Review 3

Summary and Contributions: This paper proposes Group Contextual Encoding to address the issue of lacking global context modeling in point cloud processing models. The authors borrow the similar idea of adding 2d contexture encoding into 3d and address the feature correlation issue due to data sparsity by dividing the channel into groups. They show good empirical results on SUN-RGBD and ScanNet benchmarks and conduct nice ablation studies to vindicate they claim.

Strengths: 1) the paper is well written; 2) first to extend the contextual encoding to 3d and obtain good results; simple and effective; 3) identify the data sparsity issue and proposed grouping as a solution; 4) good empirical results and nice ablation;

Weaknesses: 1) need more analysis of group number G and code word number K: there are ablation on G and K separately, but it would be nice to include experiments on varying G and K (fix C while using different combinations of G and K). This is the key to vindicate that as G gets larger, the performance of GCE will not quickly saturate as K increase. 2) did not compare with ImVoteNet [1] on SUN RGB-D V1 benchmark, but I think this can be regarded as a concurrent work. 3) it seems the gain on ScanNet result is pretty small. How large can the variance be? [1] ImVoteNet: Boosting 3D Object Detection in Point Clouds With Image Votes. CVPR 2020 https://openaccess.thecvf.com/content_CVPR_2020/html/Qi_ImVoteNet_Boosting_3D_Object_Detection_in_Point_Clouds_With_Image_CVPR_2020_paper.html

Correctness: Yes.

Clarity: Yes, the paper is pretty well written and easy to read.

Relation to Prior Work: Nice related works section.

Reproducibility: Yes

Additional Feedback: question: 1) could you explain more on why when G = 1 and K = 0, the layer is degenerated into global average pooling? typo: line 158: should be "C should be a positive integer divisible by G." instead ---- Post rebuttal feedback: Most of my concerns are addressed by the authors' feedback, and I decide to keep current rating.


Review 4

Summary and Contributions: This paper proposes a new technic to group and integrate the contextual information of point clouds.

Strengths: 1. The channel-wise based contextual grouping is somewhat novel. 2. Well organized and easy to follow.

Weaknesses: 1. The idea of grouping contextual information using a VLAD/NetVLAD like technics is not novel enough, although the channel-wise practice may have some trivial difference. How does the author compare the proposed methods with counterparts like PointNetVLAD? In the reviewer’s opinion, the only key difference between the proposed method and the PointNetVLAD is the channel-wise grouping practice, of which the contribution is not enough to be presented in NIPs. 2. There are many other methods performed much better that the proposed method in this paper on ScanNetv2 dataset. Why the author only compared with two methods in this paper (i.e. 3DSIS and VoteNet)? In the reviewer’s opinion, if the effectiveness of this method can only be proved by comparing with the “contextual grouping” based counterparts, instead of achieving the SOTA performance, then the contribution of this paper is limited. Moreover, the reviewer would like to know if this method can be used in the current SOTA method on ScanNet leaderboard to further improved their performance? 3. The author claims that the proposed group contextual problem can solve the “data sparsity”. However, this paper does not provide enough experimental analysis on how this technic will take effect to solve this problem. Since this is one of the most important issues claimed to be solved (as described in abstract) by group contextual encoding method, a comprehensive experiment is need to prove its effectiveness.

Correctness: Yes

Clarity: Yes

Relation to Prior Work: Yes

Reproducibility: Yes

Additional Feedback: The rebuttal addressed some of my concerns, but there are still some questions remaining. The key contribution is the module for point feature learning rather than a special designation only for object detection. In point cloud research, almost all task can take use of such kind of module (e.g. classification/segmentation/detection/scene recognition), but the authors only choose the object detection and voxel classification tasks to verify its effectiveness, which have very few baseline to compare with. Therefore, my concerns lies in two-folds: 1. If the proposed Group Contextual Encoding can only serve for and yield good results on object detection tasks, then its contribution is limited. 2. If the proposed Group Contextual Encoding can be potentially used for other applications, there is not enough experiment on other tasks to verify its effectiveness. After discussing with other reviewers, I would like to change my score to borderline accept. Although in my opinion, the generalization ability of this method is still not thoroughly proved in the paper (which is my biggest concern), I would like to take the opinion from reviewer #1, that the method proposed in this paper can potentially perform well on other tasks.

[Author Response · NeurIPS 2020]

We sincerely thank all reviewers for their valuable comments and suggestions. We will improve and update the draft according to the reviews. Below we respond to specific comments and concerns.

**R1: The missing training details.** We indeed are using the same training methods for fair comparisons with existing methods. Details are given in l.20 to l.26 in the supplementary material.

**R1, R2 & R4: The novelty and contribution.** We feel encouraged that most reviewers give positive evaluations, like 'simple and effective' (**R3**) and 'well motivated and yields good results' (**R1**). It is a novel contribution because we are the first to apply this method to deal with the problems of acquiring global context in the 3D sparse point clouds. Comprehensive experiments are conducted to evaluate the effectiveness and generalizability of our method and new state of the art has been established on multiple 3D benchmarks.

**R1: Splitting up the vector & R4: Solving data sparsity.** This technique has been originally introduced in (Wu et al., 2018) and is cited in the Related Works section (l.90 to l.100). In our case, by splitting up the vectors, we expand data for the "contextual encoding layer", which solves the "data-sparsity" and facilitates the learning of the global context in the sparse 3D point clouds.

**R2: The group of the vectors.** We have conducted a comprehensive analysis of the performance w.r.t. the group number (see l.28 to l.40 and Table 2 in the supplementary for details). In our experiment, the group follows the rule of "locality" that for feature of each individual point, represented by different colors, the vectors with neighboring channels will be grouped together. We actually tried other methods such as "channel shuffle" in ShuffleNet [25], seeking to weaken the constraint of "locality", but no significant performance improvement is gained.

**R3: More results for $G$ and $K$.** The performance of the original Encoding layer ($G = 1$) will saturate quickly with the code words, and the performance depends entirely on the code-word number $K$. In our approach, We mitigate this problem by introducing the "group contextual encoding" to boost the performance by exploiting "$G$". The results in the following Table A show that our method ($C \times 3$, $G = 2$) can lead to the increase on accuracy without saturation when the number of code words is increased up to 32. We will add more experimental analysis in the final version.

Table A: Ablation studies of SA2′ layer w.r.t. $G$ and $K$ on Sun RGB-D V1. $C$ is fixed to be $C \times 3$.

| K | 8 | 16 | 24 | 32 |
|---|---|---|---|---|
| $C \times 3$, $G = 1$ | 55.8 | 55.5 | **56.2** | 55.4 |
| $C \times 3$, $G = 2$ | 55.8 | 56.2 | 56.4 | **56.7** |

**R3: Comparison with ImVoteNet.** ImVoteNet is published very recently (after NeurIPS deadline). It is only evaluated on the benchmark of SUN-RGBD V1 while our method has been evaluated on other several benchmarks. Also, the results reported by ImVoteNet in that paper are not directly comparable to ours because it used additional RGB information while our method, only utilized the geometric information.

**R3: The improvement on ScanNet.** The details of the gain and variance w.r.t. the seed layer on ScanNet can be found in Table 4 of supplementary material. Actually, when measured on a more strict evaluation metrics of mAP @ 0.5, the improvement is 6.57 mAP, which can be found in Table 4, 5 in the paper. For the task of ScanNet Voxel labeling, the accuracy has been improved significantly by 1.5 (%) , surpassing the previous SOTA, PointWeb and LG-PointNet++.

**R3: Global Average Pooling.** We were following the EncNet [23], which uses $K = 0$ to denote the "global average pooling (GAP)". We will clarify this in the final version.

**R4: PointNetVLAD.** Our method is extended from the Encoding Layer [23, 24] instead of the NetVLAD [1]. The difference and the advantage of Encoding Layer [23, 24] have been clarified in [23, 24] that the encoding weight of the Encoding Layer is based on the residual. While in the NetVLAD, the weight is solely based on the input instead of the dictionary. As s result, the code words of NetVLAD are not learned from the distribution of the descriptors, making it inferior to the Encoding Layer [23, 24]. Therefore, we chose Encoding Layer as our targeting baseline. We will add a comparison with the PointNetVLAD in the final version. The concerns about the novelty and contribution regarding our method have been answered in a previous question.

**R4: Comparison with SOTA on ScanNetV2 .** It should be noted that the results in the paper are for 3D detection rather than segmentation, please refer to recent work of VoteNet [12][1]. In Table 6, 7 of [12] , only the results of VoteNet and 3DSIS are listed for 3D detection on ScanNetV2. These are the only two recent reported "Geo Only" results on this task and we included them all.

**R4: Experiments to verify the effectiveness.** To verify the effectiveness of "Grouping", we have conducted ablation experiments to study specifically the impact of "grouping". Details can be found in l.251 to l.259 with the title "Comparison with Encoding layer" and Table 2 in the paper. The results has showed that our method has outperformed the non-grouping counterparts, thus verifying the effectiveness of our "Grouping".

## Footnotes

[1] https://arxiv.org/pdf/1904.09664.pdf


[Meta-Review · NeurIPS 2020]

This submission proposes a method for processing point clouds augmented with incorporating global context features in learning. It initially received four reviews with mixed positive and negative scores (7,5,6,2). The reviewers mention excellent performance, thorough evaluation, and novelty in applying and extending known technics to object detection in 3D. The rebuttal clarified the majority of other concerns, which resulted in an increase of scores to (7,6,6,6). For these reasons, the AC's recommendation is to accept this submission as a poster.